# Targeting MicroRNAs with Small Molecules

**DOI:** 10.3390/ncrna10020017

**Published:** 2024-03-14

**Authors:** Kisanet Tadesse, Raphael I. Benhamou

**Affiliations:** The Institute for Drug Research of the School of Pharmacy, Faculty of Medicine, The Hebrew University of Jerusalem, Jerusalem 91120, Israel

**Keywords:** microRNA, therapeutics, small molecules, RIBOTAC, targeted degradation

## Abstract

MicroRNAs (miRs) have been implicated in numerous diseases, presenting an attractive target for the development of novel therapeutics. The various regulatory roles of miRs in cellular processes underscore the need for precise strategies. Recent advances in RNA research offer hope by enabling the identification of small molecules capable of selectively targeting specific disease-associated miRs. This understanding paves the way for developing small molecules that can modulate the activity of disease-associated miRs. Herein, we discuss the progress made in the field of drug discovery processes, transforming the landscape of miR-targeted therapeutics by small molecules. By leveraging various approaches, researchers can systematically identify compounds to modulate miR function, providing a more potent intervention either by inhibiting or degrading miRs. The implementation of these multidisciplinary approaches bears the potential to revolutionize treatments for diverse diseases, signifying a significant stride towards the targeting of miRs by precision medicine.

## 1. Introduction

MicroRNAs, or miRs, constitute a class of small, noncoding RNA molecules crucial for regulating gene expression within cells. Typically comprising around 22 nucleotides, these small RNA molecules play a pivotal role in the intricate processes that govern cellular gene expression [1,2,3]. Their impact on various biological processes is substantial, as they interact with messenger RNA (mRNA), either hindering the translation of mRNA into proteins or promoting its degradation [4,5]. Recognized for their precision in fine-tuning gene expression, miRs influence cell development, differentiation, and responses to diverse environmental and physiological conditions [6,7,8]. Serving as integral components in complex regulatory networks that uphold cellular homeostasis, miRs are implicated in a broad spectrum of diseases, including cancer [9,10], cardiovascular disorders [11], and neurological conditions [12,13,14]. Consequently, miRs have become a focal point in extensive research within the realms of molecular biology and medicine [15].

MiR biogenesis constitutes a complex molecular process commencing with the transcription of miR genes by RNA polymerase II [16,17,18]. This results in the generation of primary transcripts known as primary-miRs (pri-miRs), which manifest one or more hairpin structures (Figure 1). After transcription, the enzyme Drosha, in a complex with its cofactor DGCR8, orchestrates the processing of pri-miRs, yielding shorter hairpin structures referred to as precursor-miRs (pre-miRs) [18,19,20,21]. Following this processing step in the nucleus, the pre-miR is transported to the cytoplasm via exportin-5 [22,23,24]. Upon reaching the cytoplasm, the pre-miR undergoes further enzymatic cleavage facilitated by Dicer [25,26] resulting in the formation of a double-stranded RNA duplex typically comprising 21–22 nucleotides [27]. Subsequently, a strand of this duplex, denoted as the guide strand, is selectively chosen and incorporated into the RNA-induced silencing complex (RISC) [28,29,30,31]. The mature miR (mat-miR), guided by the RISC complex, engages in base-pairing interactions with target mRNAs [30,32]. This molecular interaction event results in the translational repression or enhancement of target mRNAs, thereby effecting the regulatory modulation of gene expression [30].

MiRs play diverse roles in human health and pathology, being implicated in a broad spectrum of diseases. Bioinformatics and cloning investigations suggest that miRs potentially govern around 30% of the entire human genome, with each miR capable of regulating hundreds of gene targets [33]. Dysregulation of miR expression is a frequent occurrence and in various cancer cells specific miRs are observed to be either upregulated or downregulated [34]. The overexpression of certain miRs can play a role in the onset and advancement of cancer by impacting essential cellular processes such as proliferation, apoptosis, and metastasis. Many miRs are upregulated in cancer, including miR-21, miR-155, miR-10b, miR-17-92, miR-221, miR-222 and miR-224 [35]. These miRs are only a small subset of the miRs that are upregulated in diseases. Notably, one of the most studied miRs is miR-21, which plays a pivotal role in promoting tumor growth and invasion through multifaceted mechanisms [36]. MiR-21 orchestrates a complex network of molecular events that collectively contribute to the uncontrolled growth, survival, and metastatic potential of cancer cells by upregulating genes involved in cell proliferation, inhibiting apoptosis, enhancing invasive capabilities, promoting angiogenesis, inducing epithelial-mesenchymal transition, and interacting with tumor suppressor genes [37,38]. In the case of brain tumors and pancreatic cancer, miR-21, miR-221 [39], and miR-155 [40] are overexpressed. Moreover, miR-155 is linked to breast cancer progression [41]. Additionally, miR-155 and miR-146a are dysregulated in rheumatoid arthritis [42]. Prostate cancer is characterized by an alteration in the expression levels of miR-21 and miR-141 [43]. In the case of multiple sclerosis, the autoimmune disorder involves the upregulation of miR-326 [44]. Conversely, certain miRs exhibit downregulation in many diseases, especially in cancer, including miR-34a, miR-122, miR-15, miR-16, miR-200, miR-126, miR-143, and miR-145. Notably, miR-34a, miR-122, miR-15, and miR-16 function as tumor suppressors in lung [45], liver [46], and leukemia [47] cancers. MiR-143 and miR-145 are downregulated in colorectal cancer, while the miR-200 family plays a role in regulating epithelial–mesenchymal transition in ovarian cancer [48]. MiR-1 and miR-133 are involved in cardiac hypertrophy [49,50], whereas miR-208a is associated with heart failure [51]. Alzheimer's disease involves the regulation of amyloid precursor protein processing by miR-29 [52] and in Parkinson's disease there is a downregulation of miR-133b [52]. Metabolic diseases, such as diabetes, are influenced by miR-375, which contributes to the regulation of pancreatic β-cell function [53]. MiR-2861 is associated with the regulation of osteogenic differentiation in osteoporosis [54]. Additionally, miR-122 supports the replication of the hepatitis C virus [55] and miR-28, miR-125b, and miR-150 are implicated in HIV replication [56]. Gaining a comprehensive understanding of the overexpression and underexpression of these miRs offers valuable perspectives on the development and advancement of specific malignancies, potentially revealing promising therapeutic targets for intervention.

RNA offers extensive possibilities as a central target for both small-molecule chemical probes and antisense oligonucleotide therapeutic agents [57]. It exhibits well-defined structures that can be precisely targeted by specific small molecules [58]. To effectively target RNA with small molecules, a comprehensive assessment is required, considering factors such as the uniqueness of the target site, the presence of on- and off-targets within functional regions, expression levels, and RNA turnover speed [59]. However, RNAs, with defined structures, can be precisely targeted by specific small molecules [58]. Ligand recognition in RNA structured sites aids in identifying off-targets and anticipating bioactive interactions, facilitating the determination of RNA target sequences with a preference for small-molecule binding [60,61]. Researchers are actively exploring the potential of small molecules to target RNA for therapeutic purposes. These molecules can influence RNA structures through different mechanisms, such as splicing control, translation inhibition, and deactivation of noncoding RNA functional structures [60,62,63,64,65,66,67]. Ongoing efforts involve identifying, validating, and optimizing these small molecules to advance their capacities to treat cancer and other diseases [68,69]. Recent studies have challenged the previously held belief that selectively recognizing RNA with small molecules is difficult, showing that these compounds can have selective effects on both the transcriptome and proteome [57,70,71,72,73]. Various methods, including fluorescence-based assays and dynamic combinatorial screening, are used to discover small molecules that bind to RNA structures [74].

In this review, we will discuss the biological mechanisms involved in the processing of miR, as well as various approaches employed for the identification of small molecules that target miRs, including motifs with specific binding capabilities to these molecules. Additionally, we will elaborate on strategies employed to inhibit and degrade miRs, exploring potential future developments that hold promise for targeting miRs.

## 2. Identification of Small Molecules Interacting with miRs

### 2.1. Bioinformatics Tools

Bioinformatics tools play a vital role in identifying small molecules targeting miRs by predicting miR targets, identifying small molecules targeting RNA, performing integrative analysis of miR, developing new computational algorithms, and validating potential targets [75,76,77,78]. These techniques help researchers to better understand the effects of small molecules on miR expression and function, with the goal of developing new therapeutic strategies [79]. Different discoveries with bioinformatics tools revealed the significance of bioinformatics for miR/RNA–small molecule identification [80,81,82]. A research group led by Yang developed a novel method called RNAsite to predict small molecule–RNA binding sites using sequence profiles and structure-based descriptors. This method showed the influence of the small molecule on the stability of targeted RNA [61]. Another research group introduced Harnessing RIBOnucleic acid—Small molecule Structures (HARIBOSS), designed to enable the investigation of drug-like compounds binding to RNA through techniques such as X-ray crystallography, nuclear magnetic resonance spectroscopy, and cryoelectron microscopy. The database supports the analysis of ligand and pocket properties, fostering the development of in silico approaches for the identification of small molecules targeting RNA [83]. Herein, we are discussing major bioinformatics tools that were utilized to find small molecules targeting miR.

#### 2.1.1. InfoRNA

InfoRNA, a computer-based bioinformatics database, encompasses the extensive collection of RNA motif–small molecule binding partners found in the scientific literature (Figure 2A) [84,85,86]. Mathew D. Disney et al. first introduced this in 2014 and it was upgraded by 2016. It enabled the precise identification of optimal targets for small molecules. It introduced a chemical similarity searching feature, along with an improved user interface, and is accessible for free through an online web server. The approach was effectively utilized on miR precursors, yielding a hit rate of 44% [87]. Through an extensive examination of all precursor miRs within the human transcriptome for potential matches with the RNA motif–small molecule database, InfoRNA investigated over 5,400,000 potential interactions. This involved analyzing motifs found in 1048 miR precursors, approximately 6850 motifs, and comparing them with the 792 RNA motif–small molecule interactions present in the database [79]. Through the integration of interactions identified by multiple laboratories, the database has now expanded in size, comprising 1936 RNA motif–small molecule interactions. This extensive dataset includes 244 distinct small molecules and 1331 motifs [84]. The motifs were systematically characterized based on dual criteria. Firstly, the targetable motifs were required to have an impact on the miR biogenesis process through interaction with either Dicer or Drosha [81,88]. Secondly, the inclusion criterion mandated that the miR itself should be either expressed in or identified as a causative factor for a particular disease [79]. InfoRNA also facilitated the discovery of small molecules and modular compounds designed with precision to target miRs. This was accomplished through the application of Two-Dimensional Combinatorial Screening (2DCS) and Structure-Activity Relationships through Sequencing (StARTS) analyses, rendering the approach entirely target-agnostic [89]. Consequently, bioactive small molecules can be systematically designed on a transcriptome-wide scale without inherent bias, showcasing the method's versatility and broad applications. Through this combinatorial method, InfoRNA successfully identified several miRs with potential binding ligands, including miR-18a, miR-210, miR-27a, miR-525, miR-515, miR-96, miR-21, miR-372, and miR-17-92 [85,90,91]. Using InfoRNA, Disney designed a small molecule that inhibits the Drosha site of pri-miR-525, miR-27a [91], miR-51 [90], miR-96 [92]. Other studies used InfoRNA to successfully discover small molecules that target and inhibit different miRs, like miR-21 and miR-17-92 [88,93,94]. An inherent drawback of InfoRNA is its reliance on computer-based processes, making it difficult to trust its outputs without the support of experimental validation.

#### 2.1.2. Docking 

Molecular docking is a computer-based method employed to forecast the interactions between small molecules and larger biomolecules, including enzymes, receptors, RNA, DNA, and various proteins [95]. This computational approach enables the anticipation of atomic-level interactions between a small molecule and a protein, offering researchers valuable insights into how these molecules behave within the binding site of a target protein. This predictive capability sheds light on the fundamental biochemical processes governing these interactions and is not limited to proteins but extends to the identification of small molecules interacting with RNA and miRs [96]. In the realm of molecular docking, researchers can leverage this method to forecast the binding affinity of small molecules to miRs, providing a comprehensive understanding of their interactions [97]. One research application involves validating the binding of predicted small molecules with oncogenic miRs [98]. Moreover, NMR (nuclear magnetic resonance) studies have facilitated the computational forecasting of dynamic conformational ensembles for miRs [70]. These ensembles can be utilized in silico for docking studies with libraries of small molecules, enabling specific screening for binding to miRs [70]. Numerous studies have demonstrated the efficacy of molecular docking in identifying small molecules binding to specific miRs. For instance, Duca et al. employed molecular docking to identify a molecule binding to the miR-372 Dicer site, showcasing improved inhibitory activity compared to a known binder targeting the same site [99]. In another study, Duca et al. conducted a comprehensive assessment of polyamine analogues, guided by molecular docking with the PA-1/pre-miR-372 complex. The theoretical measurements, complemented by experimental foot printing, revealed key hydrogen bonds and electrostatic interactions at specific residues of pre-miR-372, with the molecule PA-3 showing promising effects on inhibiting miR biogenesis [100]. Further studies have employed docking to identify specific binding sites of small molecules to miR targets. Maiti et al. conducted a docking study between pre-miR-21 and aminoglycosides, with streptomycin exhibiting the strongest binding. This approach also led to the identification of U-A and A-G as binding motifs [101]. In essence, molecular docking emerges as a powerful tool with diverse applications, providing valuable insights into the world of molecular interactions between small molecules and biologically relevant macromolecules. A limitation of molecular docking is its reliance on static structures, which may not fully capture the dynamic nature of RNA structures [102]. MiRs often exhibit conformational changes and flexibility, and accurately predicting these aspects during docking poses challenges, potentially affecting the reliability of the results.

#### 2.1.3. AI

AI-based bioinformatics can also be utilized to identify small molecules binding to RNA. This approach leverages artificial intelligence algorithms to analyze data, predict interactions, and expedite the discovery of potential therapeutic compounds. AI can also be used to identify small molecule interactions with miR, which can be useful in drug discovery, disease therapy, and clinical applications. This can be employed with a variety of methods, including machine learning, deep learning, and heterogeneous network representation learning. A research team has recently developed a user-friendly web server to predict the regulation of miR expression by small molecules. They used 4132 up-regulation pairs and 3182 down-regulation pairs to construct models, employing the random forest algorithm for optimal performance. The dataset was expanded to include 1509 FDA-approved small molecules and 2236 human miRs, enhancing the web server’s capabilities for drug candidate and target prediction [98]. A deep-learning-based method has been proposed for the faster detection of associations between miRs and small molecules, relying only on continuous feature representation using auto-encoders. The researchers introduced the SM2miR dataset, serving as dependable network seeds for predicting associations between small molecules and miRs. Significantly, this approach can predict potential targets of small molecules even in the absence of known target information [103]. Li et al. introduced heterogeneous network representation learning, a computational model proposed for predicting potential small molecule–miR associations based on heterogeneous network representation learning [104]. However, it is important to note that AI-based predictions should be validated experimentally to confirm their accuracy and reliability. This method is still in its infancy but represents, for sure, the future of identifying novel ligands for targeting miRs. 

### 2.2. DNA Encoded Library (DEL) 

DEL is an extensive assortment of small-molecule compounds, each linked to a distinct DNA sequence that functions as a chemical identifier [82]. This approach is an experimental assay that enables the concurrent creation and screening of numerous compounds, facilitating the efficient discovery of molecules with specific properties or activities [105]. DELs have become pivotal in drug discovery, offering high-throughput screening capabilities and the ability to explore diverse chemical space for the identification of novel bioactive compounds [106]. Employing DELs for miR binder identification offers notable advantages that significantly enhance the overall process [107,108]. First and foremost, DELs provide an extensive and diverse chemical space with numerous compounds, thereby increasing the likelihood of discovering small molecules that can effectively bind to specific miRs [109]. Additionally, DELs enable high-throughput screening against all sorts of targets, allowing researchers to evaluate a substantial number of compounds in a single experiment and expediting the discovery of potential binders [110]. Another key feature is the integration of DNA barcoding in DELs, where each compound is associated with a unique DNA sequence. Post-screening, the amplified and sequenced DNA tags of compounds binding to RNAs facilitate efficient identification and tracking of hit compounds, simplifying the hit validation and optimization process (Figure 2B) [82]. The utilization of DELs in binder identification stands out for its diverse chemical space, high-throughput capabilities, and streamlined hit identification and tracking. In a recent study, Benhamou et al. reported that a DNA-encoded library (DEL) containing 73,728 ligands was screened against a library of 4096 RNA structures. Around 300 million potential interactions were evaluated, leading to the identification of specific ligand–RNA pairs. One ligand discovered in this process targeted a specific internal loop in pri-miR-27a, an oncogenic precursor. This DEL-derived ligand was active in cells, inhibiting pri-miR-27a processing selectively. By leveraging evolutionary principles in early drug discovery, high-affinity target-ligand interactions were identified, disrupting disease pathways in preclinical models [107]. 

### 2.3. Microarray Screening

Small molecule microarray screening is an efficient high-throughput method initially designed to examine the interaction between ligands and proteins [111]. This method is also known as two-dimensional combinatorial screening (2DCS), which simultaneously explores both RNA and ligands [112]. Disney et al. developed this technique with an immobilized aminoglycoside library and probed 81,920 interactions with a three × three nucleotide RNA internal loop library. Selected RNAs binding to aminoglycosides were identified through gel excision and statistical analysis [112]. Over time, it has been adapted to identify small molecules binding to miRs. The process involves immobilizing small molecules on a glass slide, followed by the introduction of labeled miR/RNA targets [112,113,114]. This leads to the formation of complexes between miR/RNA and small molecules, detectable through fluorescence or radioactive labeling. The resulting complexes are then analyzed to identify specific binding sites and motifs of the miR/RNA (Figure 2C). Schneekloth and colleagues utilized small molecule microarray screening to discover a small molecule that binds to the Dicer cleaving site of miR-21. The primary goal of this study was to identify the specific binding motif within a library of small molecules, thereby uncovering the direct binding and inhibitory effects on miR-21. This research demonstrates the capability of the screening method to simultaneously investigate and pinpoint specific interactions between small molecules and RNA motifs [115]. The insights gained from this approach can be employed to identify other small molecules interacting with miRs, as noted by Fan et al. [97]. In a related approach, small molecules identified through a DNA-encoded library (DEL) were selectively attached and immobilized on a microarray surface. These molecules were then exposed to a radioactively labeled RNA library, along with excess oligonucleotide competitors. The RNAs bound to each small molecule were collected and sent for sequencing to identify the binding motifs of the small molecule binder. Microarray screening provides a powerful means for high-throughput analysis, enabling biomarker discovery and gene expression profiling. However, its utilization is constrained by high costs, data analysis complexity, and the risk of cross-hybridization. 

### 2.4. Fluorescence Reporter

Florescence assays allow for a high throughput screening approach, making it possible to test a large number of small molecules for their interactions with miRs efficiently [116]. It provides a valuable tool for researchers aiming to identify and characterize compounds that can selectively bind to and influence the activity of specific miRs, which has implications in therapeutic interventions [74]. There are several ways that florescence assays can provide crucial evidence about the binding of small molecules to RNA and specifically to miRs.

#### 2.4.1. Fluorescence Indicator Displacement (FID) 

FID is a technique used to study molecular interactions, particularly in the context of sensing or detecting specific ligands [117,118]. The basic principle involves a fluorescent indicator molecule that is displaced from a binding site by a target ligand, resulting in a change in fluorescence intensity. This method is often employed in the identification of small molecules interacting with RNAs [118]. FID works by designing a fluorescent indicator molecule to bind to a specific site on an miR target (Figure 2D). Then the desired ligand, a specific miR binder molecule, is introduced into the system [119]. If the ligand has a higher affinity for the binding site than the fluorescent indicator, it competes for binding and displaces the fluorescent indicator from the binding site [120]. The displacement of the fluorescent indicator results in a change in the fluorescence signal. This change is often monitored as an increase or decrease in fluorescence intensity. This method is valuable for developing small molecules targeting miRs [121]. A study conducted by Nakatani et al. utilized this approach to discover small molecules binding to pre-miR-21. They employed a competitive binding assay based on fluorescence polarization. This method screened molecules that competed with a moderately binding ligand for pre-miR-21 [120]. In general, FID assay is a sensitive approach that can be applied to detect small molecules binding to miR. The limitation of FID lies in its dependence on the binding affinity of the indicator molecule, which can potentially hinder the accurate detection of target analytes.

#### 2.4.2. Fluorescence Resonance Energy Transfer (FRET) 

FRET is a powerful and widely used technique in molecular and cell biology to study molecular interactions and conformational changes at the nanometer scale [122]. FRET relies on the transfer of energy between two fluorophores (molecules that fluoresce) when they are in close proximity [123]. FRET involves two fluorophores, commonly referred to as the donor and acceptor. The donor fluorophore absorbs light at a certain wavelength and emits light at a longer wavelength [124]. The acceptor fluorophore then absorbs this emitted light when the donor fluorophore is in close proximity to the acceptor fluorophore (typically within a range of 1 to 10 nanometers). The energy from the excited donor is transferred to the acceptor. This transfer occurs through non-radiative dipole–dipole interactions [123]. The energy transfer results in a decrease in fluorescence from the donor and an increase in fluorescence from the acceptor (Figure 2E) [125]. By measuring changes in fluorescence intensity or lifetime, researchers can infer the proximity and interaction between the two fluorophores [126]. FRET is commonly used to study protein–protein interactions, conformational changes, and molecular dynamics in living cells [127]. In the context of RNAs or miRs, FRET probes can be designed to study the hybridization or interactions between specific sequences [128,129]. It can also show the real-time interaction between a small molecule and a given miR [130]. Duca et al performed FRET-based assays to describe the inhibition activity of small molecules on pre-miR-21 [131,132]. In summary, FRET is a versatile and sensitive tool that has broad applications in molecular and cellular biology, including the study of miR interactions [133,134,135]. FRET offers sensitive, real-time insight into molecular interactions within living cells. However, challenges such as background noise and distance constraints between fluorophores can affect data quality.

## 3. Inhibition of miR Processing

The inhibition of miR biogenesis involves disrupting the various stages of miR production to regulate cellular processes and gene expression [5]. Several strategies can be employed to inhibit miR biogenesis, and these methods target key components of the miR processing pathway [136,137]. MiR biogenesis is a multistep process which involves the transcription of miR genes, the generation of precursor miRs, and their subsequent maturation into functional miRs. The key stages include the transcription of pri-miR by RNA polymerase II, processing of pri-miR into pre-miR by the Drosha-DGCR8 complex in the nucleus, transport of pre-miR to the cytoplasm, and its further processing by Dicer to form mature miR duplexes. Finally, one strand of the mature miR is loaded into the RNA-induced silencing complex (RISC), guiding the complex to target messenger RNAs (mRNAs) for post-transcriptional regulation (Figure 1). Several strategies can be employed to inhibit miR biogenesis, and these methods target key components of the miR processing pathways, such as Drosha and Dicer cleaving sites [133]. This can be achieved by designing small molecules that specifically bind to the active sites of Drosha and Dicer to disrupt their enzymatic activity. These inhibitors could prevent the cleavage of precursor miRs, halting the maturation process [138].

### 3.1. Inhibition of the Drosha Site 

Inhibiting the biogenesis of miRs necessitates a focus on targeting crucial processing sites. Numerous studies underscore the significance of hindering miR processing at the Drosha site, as it exerts a pivotal influence on the overall inhibition of miR biogenesis [134]. Various approaches have been employed to target Drosha, including the inhibition of the Drosha-DGCR8 complex. This complex requires flanking nonstructured RNA sequences for efficient processing [138]. The importance of these diverse methodologies has been highlighted in distinct research studies, emphasizing the critical role of targeting Drosha for effective modulation of miR biogenesis (Figure 3) [139,140]. 

The use of a small molecule to inhibit Drosha has demonstrated effective diminution of miR maturation (Figure 3) [135,141]. A number of studies have utilized different approaches to target and inhibit the Drosha functional site (Table 1). Benhamou et al. discovered a small molecule that inhibits pre miR-27a at the Drosha functional site. In this article, they used a DEL with 73,728 ligands and incubated them with two distinctively labeled RNA members with 4096. Sequencing and informatics analysis unveiled affinity landscapes and potential target miRs for each identified ligand. A compound with nanomolar affinity for oncogenic pri-miR-27a was discovered, exhibiting the ability to impede miR biogenesis by binding to the A-C internal loop and rescue a migratory phenotype in triple-negative breast cancer cells [107]. Notably, this team discovered another small molecule targeting a Drosha site. In this study, they used InfoRNA to uncover the ligandable oncogenic miR target which is pri-miR-27a and synthesized a homo-dimeric small molecule that can target the Drosha and its adjacent internal loop. The small molecule effectively inhibited pri-miR-27a biogenesis in two breast cancer cell lines, MDA-MB-231 and MCF-7, as well as in the prostate cancer cell line LNCaP [91]. Disney et al. used 2DCS to discover small molecules targeting a specific motif (A-A internal loop) in Drosha of miR-10b. The compounds, identified as guanidinylated kanamycin A (G Kan A) and guanidinylated neomycin B (G Neo B), were found to lead to a reduction in the levels of mature miR-10b [109]. MiR-10b is known to directly inhibit the mRNA of HomeoboxD10 (HOXD10), a tumor suppressor that hinders genes associated with cell migration and extracellular matrix remodeling, including RhoC, a3-integrin, and Mt1-MMP. To investigate this, the researchers evaluated whether G Neo B could relieve the repression of downstream targets of miR-10b in a model system, showing an increase in the expression of proteins regulated by miR-10b. In another study the team utilized InfoRNA to discover a homo-dimeric small molecule named Targaprimir-515/885. This molecule was designed to bind to motifs located in the Drosha processing sites of miR-515 hairpin precursors specifically. The outcome of the study demonstrated the inhibitory effect of Targaprimir-515 on the biogenesis of miR-515 (Table 1) [90]. This dimer small molecule was identified to target two adjacent hairpin precursors of the miR-515 Drosha site to inhibit its biogenesis [90]. Through InfoRNA the authors identified a bis-benzimidazole molecule that binds to a 1 nt × 1 nt internal loop near a Drosha site, resulting in increased sensitivity of cells to Herceptin. This finding was validated in two additional cell lines, HepG2 and MDA-MB-231, both of which express miR-515 and lack HER2. Notably, normal breast epithelial cells (MCF-10A), which do not express miR-515, remained unaffected by this treatment. Disney et al., in another study, targeted miR-96 using a hetero-dimer small molecule named Targaprimir 96 [92]. This molecule selectively modulated miR-96 and induced apoptosis. The compounds inhibited the biogenesis of miR-96 by binding to the Drosha and its adjacent internal loop motifs (U-U and G-G, respectively). In a related study from the same group, a small molecule targeting pri-miR-525 was identified using InfoRNA. This molecule specifically binds to the U-C internal loop of pre-miR-525. Consequently, 5'-azido neomycin B was found to decrease the levels of mature miR-525, enhance ZNF395 expression, and inhibit the invasiveness associated with the oncogenic miR-525 [87]. 

### 3.2. Inhibition of Dicer Site

Various methods can be employed to inhibit Dicer activity, thereby preventing miR processing. These approaches include proximity-enabled Dicer inactivation, l-RNA aptamers, specific proteolysis inhibition, and modulation of Dicer activity through associated dsRNA binding proteins [133,142,143,144]. The inhibition of the functional site of Dicer represents a common strategy to block miR processing (Figure 3). Numerous studies have explored small molecules targeting the Dicer functional site, showcasing innovative and diverse methodologies. Significantly, a considerable number of these compounds were designed to target miR-21, a highly oncogenic miR that is frequently overexpressed in various types of cancer. Yan et al. utilized bifunctional molecules, incorporating neomycin as a binder for pre-miR-21 at the U-U internal loop motif and N-hydroxyimide as a Dicer inhibitor (Table 2) [121]. The binder molecule was identified by FID. This approach successfully inhibited miR-21 biogenesis [121]. Varani et al. also discovered drug-like small molecules that bind specifically to the precursor of the oncogenic miR-21. The small molecules target a local structure at the Dicer cleavage site and induce distinctive structural changes in the RNA, which correlate with specific inhibition of miR processing [145]. The compound led to reduced cell proliferation in AGS and ASPC1 cells and elicited RNA-specific responses, causing a decrease in mature miR-21 levels and the restoration of the tumor suppressor PDCD4 [145]. In another study, through a series of in vitro and cell-based assays, Bose et al. found Streptomycin to effectively suppress miR-21 levels by directly binding to its precursor form, interfering with downstream processing by Dicer [101]. Notably, Duca et al. conducted biochemical studies and molecular docking to create enhanced conjugates targeting the Dicer site of miR-21 (Table 2). The results demonstrated improved affinity and efficacy in inhibiting miR-21 [132]. Moreover, Duca et al. recently introduced a novel approach involved de novo design of an RNA binder with a dihydropyrrolopyridine scaffold, resulting in ligands with increased selectivity and affinity for targeting miR-21. These compounds demonstrated efficacy in inhibiting miR-21 biogenesis [110]. Polyamine molecules, applied by Duca et al., demonstrated high affinity and selectivity for the cleaving site of pre-miR-372 (U-G internal loop), thus leading to the inhibition of the production of the corresponding miR. Some of these compounds showed an antiproliferative effect which is highly specific for gastric cancer cells overexpressing the targeted miR-372 and miR-373 [131]. Furthermore, the same research team discovered the application of polyamine molecules in targeting and inhibiting miR-372 through molecular docking. Within this group of compounds, PA-3 emerged as the most promising, exhibiting strong affinity, sub-micromolar inhibitory activity, and selectivity for miR-372 [100]. Shi et al. through screening compound libraries discovered AC1MMYR2, a small-molecule inhibitor directly suppressing Dicer's processing of pre-miR-21 into mature miR-21, effectively inhibiting tumor growth and invasion [146]. Naro et al. screened over 300,000 compounds resulting in the discovery of a novel class of small-molecule inhibitors for miR-21. Further studies on the structure–activity relationship yielded four potent inhibitors with selectivity toward miR-21 over other miRs. These compounds interact with the functional site of Dicer by operating downstream of miR transcription [147]. Ankenbruck et al., through extensive structure–activity relationship (SAR) studies, identified ether-amide inhibitors, discovered through the screening of over 300,000 small molecules, exhibited potent inhibition of miR-21 biogenesis by interacting with the functional site of Dicer [148]. Schneekloth et al. utilized a small molecule microarray (SMM) screen on an extensive library of drug-like molecules to discover new motifs binding to the pre-miR-21 hairpin. Compounds that exhibited selective binding to miR-21 were compared to other oligonucleotides, validating their inhibition activity on the Dicer functional site (Table 2) [115]. Mathew D. Disney et al. have contributed to most of the molecules discovered and developed to inhibit the Dicer site of miR processing. In one of their studies, they used InfoRNA to investigate Targapre-mir-210, a bis-benzimidazole small molecule which binds to the Dicer processing site of miR-210, regulating the miR-210 hypoxic circuit in breast cancer cells [149]. They further identified 2-Dioxy streptamine conjugates and nucleobase-conjugated series that showed promising binding and inhibition of pre-miR-372, outperforming neomycin counterparts [99]. Disney et al. employed diverse methodologies to identify natural products (NPs) and their miR-binding partners, using a two-dimensional combinatorial screening (2DCS) approach. Interactions were integrated into InfoRNA, guiding the selection of lead small molecules for RNA targets. Notably, NOC-1 was identified for its binding to the Dicer site of miR-18a and inhibiting its biogenesis [150]. In this study, they developed a hetero-dimer small molecule capable of simultaneously binding to two structural elements in pre-miR-200c, including the Dicer processing site (U-U internal loop, Table 2). The interaction with this small molecule inhibits Dicer processing of pre-miR-200c, leading to a decrease in mature miR-200c levels and subsequent reduction in cell apoptosis, which is a characteristic of the disease phenotype [141]. Overall, our analysis underscores a preference for inhibiting miR processing at the Dicer site rather than the Drosha site, likely driven by evidence suggesting that miR-21 features a binding site more readily inhibited next to the Dicer cleavage site. Additionally, we note a lower occurrence of dimeric compounds in comparison to Drosha inhibition.

## 4. Degradation of miR

Suppression of miR processing can occur by binding to either Drosha or Dicer sites. However, alternative methods to hinder miR biogenesis exist, with degradation being a key mechanism. Degradation can target different sites, functional or nonfunctional. MiRs are vulnerable to degradation through two distinct methods: bifunctional small molecules, like RIBOTAC, and direct cleaving, employing chemical agents that disrupt ribonucleic bonds directly. These strategies offer diverse ways to manipulate miR levels and function, contributing to a nuanced understanding and potential modulation of miR regulatory networks [151,152,153,154].

### 4.1. Ribonuclease-Targeting Chimera (RIBOTAC) 

RIBOTAC is a term that refers to a novel class of small molecules designed for the targeted degradation of RNA. These molecules are engineered to selectively recognize and bind to specific RNA sequences, leading to the degradation of the targeted RNA [154]. The primary breakthrough involved the transformation of RNA-binding molecules into RNA-degrading molecules. Disney and colleagues introduced the RIBOTAC concept by connecting RNA-binding molecules with a small compound that attaches to and triggers ribonuclease L (RNase L) (Figure 4A) [149]. The development of RIBOTACs for targeting disease-associated RNA or miRs has been proven to be an interesting approach, which aims to manipulate gene expression at the miR level for therapeutic purposes. RIBOTACs represent a promising approach for modulating gene expression by inducing the degradation of disease-related miR molecules [151,155]. Disney and colleagues developed a small molecule, TGP-210, identified via InfoRNA, and transformed it into a RIBOTAC by linking it to an RNase L recruiter. Through a fluorescence-based assay, they characterized its binding. TGP-210 effectively induced degradation of miR-210 in hypoxic cancers by specifically binding to the C-C internal loop [156]. In a recently published investigation, they developed a RIBOTAC molecule designed to target the nonfunctional region of pre-miR-155. The small molecule precisely binds to a nonfunctional site on pre-miR-155 and activates the recruitment of RNase L, causing substantial cleavage and resulting in a significant reduction in both precursor and mature levels of miR-155. This suggests that the degradation strategy can be implemented with small molecules that have no biological outcome since they are situated in non-processing binding sites. This offers a versatile approach that can turn a silent binder into an active degrader (Table 3) [152]. In a separate investigation, Dovitinib underwent a rational reprogramming approach, where it was strategically repurposed for targeting pre-miR-21. The RIBOTAC chimeric approach switched the affinity of the new molecule toward miR-21 instead of its original kinase target. This compound, in addition to leveraging Dovitinib’s properties, also recruited RNase L to catalytically induce the degradation of the targeted miR-21. The study demonstrated the successful degradation of miR-21, achieved through the specific binding to the Dicer site within the C-C internal loop (Table 3) [93]. In a second study, using the same Dovitinib RNA binding molecule, they modified the RNase L recruiter molecule into a new molecule identified via DEL screening [157]. A homo-dimer molecule was identified by InfoRNA and bound to the A-U, A-U internal loop in pre-miR-21. This dimer was converted into a RIBOTAC molecule (Table 3) [94]. This conversion led to the targeted degradation of the oncogenic pre-miR-21 and a decrease in mature miR-21. InfoRNA also identified a hetero-dimer small molecule that bound to the G-U internal loop using the first ligand, the U-U internal loop using the second ligand, and to the pre-miR-96. By connecting it to an RNase L recruiter moiety, the RIBOTAC molecule was engineered with the aim of inducing miR-96 degradation [158]. Overall, the RIBOTAC approach is opening a wide range of opportunities to affect miRs and many other RNA targets.

### 4.2. Direct miR Degradation 

#### 4.2.1. Bleomycin

The bleomycins (BLMs), initially isolated from *Streptomyces verticillus*, are a family of complex antitumor antibiotics [163]. Since their discovery in the mid-1960s, these glycopeptide-derived compounds, primarily BLM A2 and BLM B2, have been extensively studied and used clinically to treat various cancers, such as non-Hodgkin’s lymphoma, squamous cell carcinomas, and testicular tumors [164]. The therapeutic effectiveness of BLMs is attributed to their capacity to bind to cellular DNA, leading to oxidative degradation, particularly in the presence of certain metal ion cofactors, notably iron [153,165,166]. Recent studies showed that BLMs also have the power to affect and cleave RNAs (Figure 4B) [167]. Notably, in a study published in 2023, the Disney group compared a pre-miR-372 RIBOTAC degrader with pre-miR-372 BLM degrader and found that the BLM degrader selectively degrades the miR-372 by binding to A-bulge, while having no effect on m-JUN [88]. In a second study a hetero-dimer small molecule binding to the Drosha site of pri-miR-96 was connected with bleomycin, resulting in increased induction of apoptosis in triple negative breast cancer [160]. Notably, the hetero-dimer molecule binds to the G-G and U-U internal loop of the pri-miR-96 (Table 3). In a different investigation, the same researchers employed InfoRNA to create a ligand specifically designed to target the pri-miR-17-92 cluster, a direct transcriptional target regulated by c-MYC. They achieved enhanced potency, approximately tenfold, by directly and oxidatively cleaving the cluster, using a conjugation of bleomycin A5, to their lead compound. The resulting compound effectively degraded the entire miR-17-92 cluster [161]. Duca et al. recently introduced a new group of small molecules linked with BLM which showed a better binding and inhibition of miR-21. A molecular docking study indicates that the potential binding site is specifically located in the cleavage site of Dicer, distinguishing it from other effective binders that target different regions without inducing the same inhibitory activity. To validate this potential, the group assessed the biological activity on glioblastoma U87 cells overexpressing miR-21, miR-18a, and miR-148a. Two compounds showed notable anti-proliferative activity, particularly against cancer cells, sparing healthy cells. Despite moderate activity, these compounds show promise in impacting the proliferation of glioblastoma cells and cancer stem cells, offering a potential avenue for discovering new anti-cancer molecules for this aggressive and incurable brain cancer [159].

#### 4.2.2. Direct Degradation via Alternative Cleaving Agents 

There are various molecules which can be utilized to cause degradation in different ways when connected with targeting moiety. In a study published in 2023, Zhang et al. used a fluorophore molecule to degrade oncogenic miR-210 [162]. The chimeric molecule that they generated, TGP-210-Ppa, composed of the photosensitizer pyro pheophorbide a (Ppa) linked to the ligand of the oncogenic precursor miR-210, exhibited specific binding to oncogenic pre-miR (Table 3). Upon exposure to red light, it generated singlet oxygen, a reactive oxygen species, leading to the degradation of the target pre-miR. This innovative modification of precursor miR, through a bifunctional chimera, offers a distinctive approach to the regulation of target genes with temporal–spatial precision facilitated by photo-irradiation. The study demonstrated that TGP-210-Ppa effectively hindered the production of functional miR-210 in breast cancer cells in a photo-controllable manner. Furthermore, this intervention successfully reversed the downstream oncogenic signaling pathway activated by miR-210, ultimately promoting the death of cancer cells [162].

## 5. Prospects in Targeting miR with Small Molecules 

As outlined in this review, the predominant focus in therapeutic research involves the post-transcriptional gene silencing of targeted miR. However, this emphasis represents just one facet of miR function and biogenesis. Exploring studies on various miR-modulated biological processes could unveil innovative therapeutic targets beyond mRNA silencing. Recent investigations into targeting diverse sites of miR, both functional and nonfunctional, suggest that breakthrough methods for treating cancer and other miR-related diseases through targeted degradation therapies may emerge. This is crucial, as it implies that degrading miR biogenesis may not necessarily require mandatory targeting of miR functional sites. Advancing our understanding in this direction necessitates further studies, particularly on the identification of small molecule binders. According to the studies reviewed here, small molecule inhibitors can effectively target specific miRs, leading to significant biological outcomes that ultimately result in the demise of disease. 

### 5.1. Comparison of ASOs vs. Small Molecules 

The selection between antisense oligonucleotides (ASOs) and small molecules for therapeutic treatment through miR inhibition necessitates a nuanced evaluation of each approach's distinctive attributes. ASOs are designed to target miRs function by binding to these molecules, obstructing their activity and subsequent impacts on gene expression. ASOs present a precise avenue for modulating specific miR activities, offering the potential for personalized therapy tailored to the distinct miR profiles associated with various diseases. Nevertheless, persistent challenges, such as efficient delivery to target tissues and the risk of off-target effects, remain unresolved [168]. Moreover, the intricate structure of miRs presents a potential obstacle to the effective recognition of antisense oligonucleotide (ASO) sequences, thereby diminishing their impact. In contrast, small molecules designed for miR inhibition target post-transcriptional sites in miR biogenesis or function. This broader approach allows for interference with multiple miR pathways, either through inhibition or degradation. Notably, the convenience of the oral administration of small molecules is significant. The field of identifying small molecules targeting miRs is rapidly advancing and offers numerous advantages over ASOs. Anticipated new discoveries in this realm are expected based on the employed approaches, alignment with therapeutic goals, and preferred delivery profiles.

### 5.2. Future Approaches to Degradation of miRs

The degradation of miRs stands as a pivotal tool for advancing small molecules that influence miRs, particularly through the RIBOTAC approach. However, this technique is still in its infancy and numerous developments are expected to enhance its impact. In addition to leveraging the RNase L enzyme for RIBOTAC degradation, alternative enzymatic pathways can be explored. For instance, recruiting XRN1 and XRN2 enzymes, known for their proficiency in cleaving miRs, represents a promising avenue for further investigation. This involves connecting the enzyme recruiter to small molecules that can be tailored to selectively identify and bind to specific RNA sequences. XRN1 and XRN2, acting as pivotal 5′-3′ exoribonucleases, play essential roles in RNA degradation, contributing significantly to gene expression regulation, RNA turnover, and mRNA decay. By selectively recruiting XRN1 and XRN2, these enzymes catalyze the removal of nucleotides, ultimately leading to the targeted degradation of RNA. An innovative alternative in recent research is the development of PINADs—proximity-induced nucleic acid degraders—inspired by ribonuclease mechanisms. Comprising a small molecule RNA binder, a flexible linker, and an imidazole (found in ribonucleases), PINADs bring imidazole into proximity with target RNA sites, offering a promising approach to RNA degradation [169]. This new strategy of degradation can be potentially applied to miR degradation in the future.

### 5.3. Potential Methods for Indirectly Increasing Depleted miRs 

Several miRs, particularly those depleted in various diseases, present potential targets for small molecules. Molecules can be potentially designed to enhance or promote the biogenesis of these miRs, offering a protective approach against disease. For instance, decreased levels of the tumor suppressor miR-107 in plasma are linked to tumor progression and unfavorable outcomes. Reinstating plasma miR-107 levels is suggested as an innovative strategy for treating prostate cancer [170]. Additionally, researchers have identified an intragenic miR, specifically miR-3614, which inhibits the expression of its host gene TRIM25. Notably, IGF2BP3 competes for the binding site, impeding miR-3614 maturation and protecting TRIM25 mRNA from degradation. Elevated levels of miR-3614-3p effectively suppress breast cancer cell growth by reducing TRIM25 expression. Small molecules like Rigosertib are used to inhibit IGF2BP3, which results in decreased TRIM25 levels and suppressed cell proliferation and shows a collaborative effect with miR-3614-3p overexpression. The let-7 miR family is renowned for its tumor-suppressor role, often downregulated in cancer cells. The Lin28 protein plays a pivotal role by binding to let-7 miR precursors, inhibiting their maturation, and consequently impairing their tumor-suppressive functions [171,172]. Inhibiting Lin28 protein using small molecules leads to elevated levels of let-7 miR, consequently inducing tumor-suppressive functions [173]. This highlights the potential for small molecule intervention in depleted miRs, offering promising results in tumor suppression. Targeting RNA binding proteins and promoting the restoration of depleted miRs is a novel way to suppress tumors and treat oncogenic conditions. This approach not only addresses the intricate network of regulatory elements within the cellular machinery but also underscores the potential to modulate key factors influencing the delicate balance between pro-tumorigenic and anti-tumorigenic processes. Strategically manipulating RNA-binding proteins, or the origin of miR repression, and facilitating the resurgence of specific miRs holds the potential to disrupt disease pathways, providing a nuanced and potentially effective therapeutic avenue. However, these potential methods involve designing compounds that do not interact directly with the RNA, in contrast to the strategies mentioned in this review.

## 6. Conclusions

A plethora of strategies are available for the targeted modulation of miRs, particularly focusing on inhibiting or degrading miRs and concurrently promoting or enhancing the biogenesis of tumor-suppressive miRs through diverse mechanisms. The ongoing advancements in this field involve the continuous development of various bioinformatics tools and methodologies intended to accelerate the identification of small molecules capable of effectively binding to disease-associated miRs. A notable approach to this pursuit involves the integration of different cleavers with binder molecules, thereby facilitating the decrease of disease associated miRs. This multifaceted landscape of targeting miRs underscores the dynamic and evolving nature of research aimed at unravelling the complexities of miR regulation in the context of diseases, particularly cancer. However, disadvantages include off-target effects due to the miRs’ ability to bind to a myriad of mRNA targets, not all of which are necessarily relevant to a specific disease. Additionally, designing small molecules with sufficient affinity and selectivity for specific miRs remains challenging, impeding their clinical translation. Understanding these issues requires rigorous computational studies and experimental validation, including in vitro assays and cellular profiling, to optimize the safety and efficacy of miR-targeting small molecules. The comprehensive exploration of these various mechanisms not only enhances our understanding of miR-related processes but also holds significant promise for the development of innovative therapeutic interventions and precision medicine strategies for diseases associated with dysregulated miR expression. 

## Figures and Tables

**Figure 1 ncrna-10-00017-f001:**
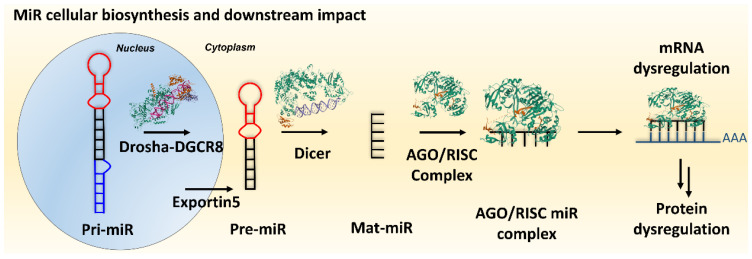
MiR processing in cells.

**Figure 2 ncrna-10-00017-f002:**
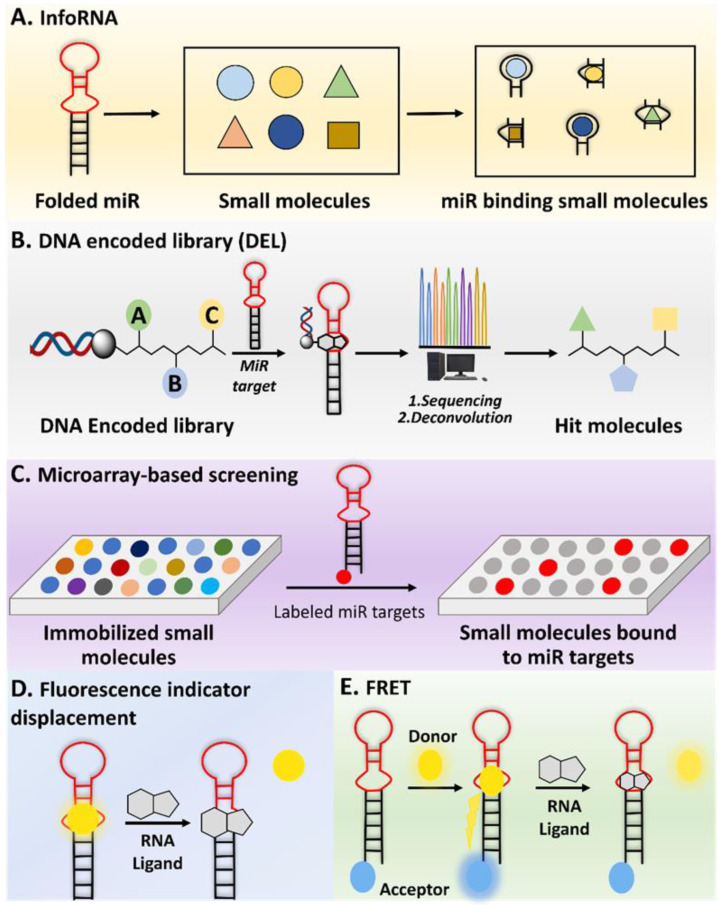
Approaches for identifying small molecules that bind to miR. (**A**) InfoRNA is a database used to identify small molecules binding to RNA. (**B**) DEL is a collection of diverse small molecules, each tagged with a unique DNA sequence, enabling high-throughput screening. (**C**) A microarray-based assay is a high-throughput experimental technique used to simultaneously evaluate the interaction between RNA motifs and small molecules. (**D**) FID is a technique where a fluorescent probe bound to a target is replaced by another molecule, causing a measurable change in fluorescence signal. (**E**) FRET is a process where energy moves between two nearby fluorophores, enabling the study of molecular interactions.

**Figure 3 ncrna-10-00017-f003:**
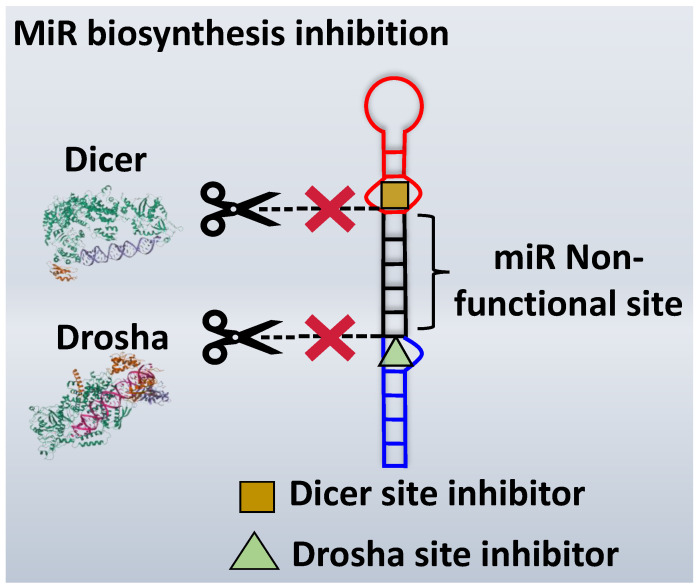
Inhibition of miR biosynthesis using small molecules. When small molecule ligands target Drosha or Dicer, it can interfere with their enzymatic activities, leading to the inhibition of miR biogenesis. This, in turn, disrupts the normal regulatory functions of miRs in controlling gene expression.

**Figure 4 ncrna-10-00017-f004:**
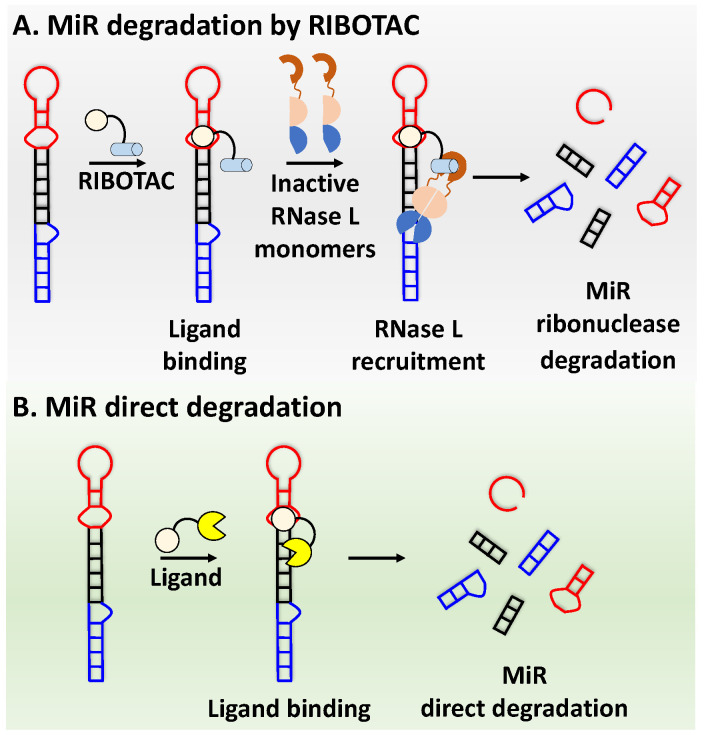
Approaches for miR degradation. (**A**) RNA degradation through RIBOTAC, and the recruitment of RNase L—a mechanism designed for targeted and efficient RNA cleavage. (**B**) MiR degradation by employing chemical agents that disrupt ribonucleic acid bonds directly.

**Table 1 ncrna-10-00017-t001:** Small molecules inhibiting Drosha-mediated miR processing. Monomer molecules are highlighted in grey, homo-dimers in green, and hetero-dimers in pink.

Pri/Pre-miRs	Small Molecules	Discovery Methods	Motifs	Ref.
Mir-27a	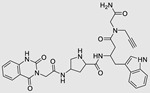	DEL	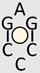	[95]
Mir-27a	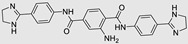	InfoRNA	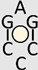	[91]
MiR-27a	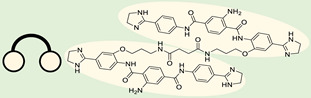 Homo-dimer	InfoRNA	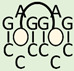	[91]
MiR-10b	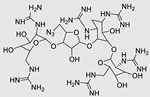	2DCS	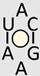	[96]
MiR-96	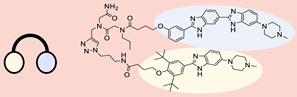 Hetero-dimer	InfoRNAChem-CLIP	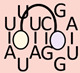	[92]
MiR-515	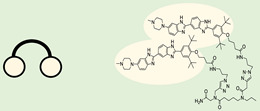 Homo-dimer	InfoRNA	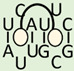	[90]
MiR-525	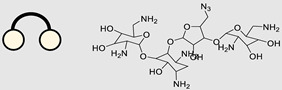	InfoRNA	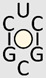	[87]

**Table 2 ncrna-10-00017-t002:** Small molecules inhibiting Dicer-mediated miR processing. Monomers are highlighted in silver and the hetero-dimer is highlighted in green.

Pri/Pr-miRs	Small Molecules	Discovery Methods	Motifs	Ref.
MiR-21	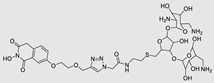	Fluorescence Polarization-Based Screening	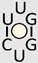	[121]
MiR-21	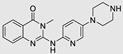	NMR Assay	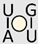	[136]
MiR-21	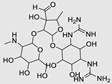	Dual Luciferase Reporter Assay	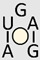	[103]
MiR-21	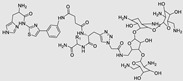	FRET	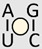	[132]
MiR-21	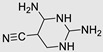	High-Throughput Screening	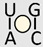	[137]
MiR-21	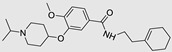	High-Throughput Screening	Not defined	[138]
MiR-21	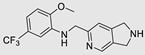	Docking	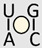	[110]
MiR-21	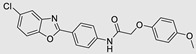	High-Throughput Screening	Not defined	[139]
MiR-21	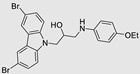	MicroArray	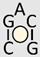	[115]
MiR-210	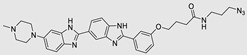	InfoRNA	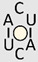	[140]
MiR-372	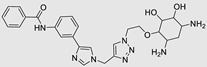	Molecular Docking	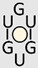	[101]
MiR-372	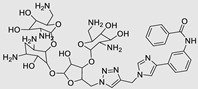	FRET	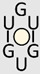	[131]
MiR-372	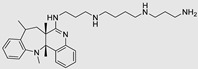	Molecular Docking	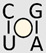	[102]
MiR-18a	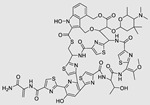	2DCS	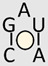	[141]
MiR-200	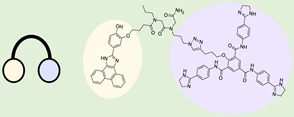 Hetero-dimer	Chem Clip	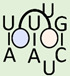	[142]

**Table 3 ncrna-10-00017-t003:** Small molecules degrading miRs. The gray background indicates RIBOTAC molecules, while the blue circles indicate the RNase L recruiter. The sky blue background shows direct degradation by bleomycin (BLM), marked in yellow, while the orange star indicates photosensitizer pyro pheophorbide a (Ppa).

Pri/Pre-miRs	Small Molecules	Discovery Methods	Motifs	Ref.
MiR-155	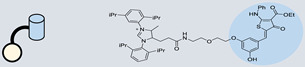	2DCS	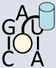	[152]
MiR-21	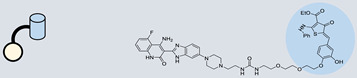	Binding assay + 2DCS	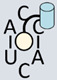	[93]
MiR-21	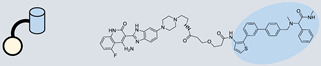	Binding assay + 2DCS	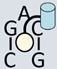	[157]
MiR-21	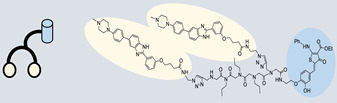	InfoRNA	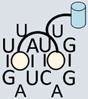	[94]
MiR-210	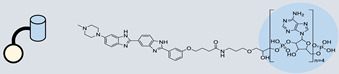	InfoRNA	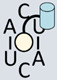	[156]
MiR-96	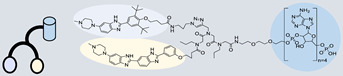	InfoRNA	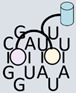	[158]
MiR-372	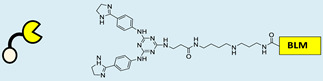	InfoRNA	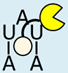	[88]
MiR-21	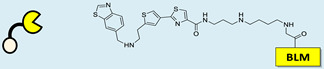	Molecular docking	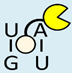	[159]
MiR-96	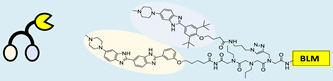	InfoRNA	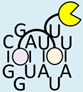	[160]
MiR-17-92	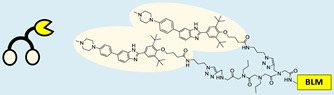	InfoRNA	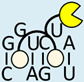	[161]
MiR-210	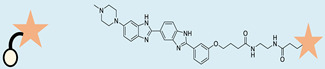	InfoRNA	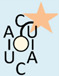	[162]

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
