# Peer review of "Targeting MicroRNAs with Small Molecules"

_ncrna, 2024, doi:10.3390/ncrna10020017_

Round 1
Reviewer 1 Report
Comments and Suggestions for Authors
In the paper entitled: "Targeting MicroRNAs with Small Molecules", the authors discuss the progress made in drug discovery processes and analyze several approaches by which researchers can systematically identify compounds to modulate miR function, providing a more potent intervention either by inhibiting or degrading miRs.
The knowledge and implementation of the different multidisciplinary approaches could improve the treatments for diverse diseases, leading toward precision medicine targeting miRs.
This review is fascinating in the field of the possible use of small molecules targeting ncRNAs playing a role in the pathogenesis of different diseases. The authors have collected and discussed a large number of literature data recently published.
Minor:
- I suggest the authors add a section about the methodologies used to collect the bibliography used for this review.
- Could the authors better explain the paragraph 2.1.3 AI by adding the extension of "AI" and their significance
Comments on the Quality of English Language
- I suggest an English language revision to improve the quality of the paper
Author Response
Reviewer one
In the paper entitled: "Targeting MicroRNAs with Small Molecules", the authors discuss the progress made in drug discovery processes and analyse several approaches by which researchers can systematically identify compounds to modulate miR function, providing a more potent intervention either by inhibiting or degrading miRs.
The knowledge and implementation of the different multidisciplinary approaches could improve the treatments for diverse diseases, leading toward precision medicine targeting miRs.
This review is fascinating in the field of the possible use of small molecules targeting ncRNAs playing a role in the pathogenesis of different diseases. The authors have collected and discussed a large number of literature data recently published.
Reviewer #1, Comment 1: I suggest the authors add a section about the methodologies used to collect the bibliography used for this review.
Response: We thank the reviewer for his/her comments. We used keywords such as microRNA targeting small molecules, computational studies for microRNA targeting ligands, RIBOTAC, direct degradation of microRNAs to help the readers understand from which topic each research was collected.
Reviewer #1, Comment 2: Could the authors better explain the paragraph 2.1.3 AI by adding the extension of "AI" and their significance.
Response: We have explained the significance of AI with references on microRNA such as the AI-based bioinformatic dataset that includes 1509 FDA-approved small molecules and 2,236 human miRs, enhancing the web server's capabilities for drug candidate and target prediction. Additionally, we also discussed another AI, the SM2miR dataset.
Reviewer 2 Report
Comments and Suggestions for Authors
This review article by Tadesse and Benhamou comprehensively describes how small molecules can be used to impair maturation, stability, and function of microRNAs, representing a valuable alternative to “classic” antisense inhibitors.
The review is well conceived and very well written. I commend the authors because it is a pleasure to read it.
These are my comments for the authors:
MAJOR
1-Please, include in the text a description of the ALIS method (panel G in Fig 2).
2-In chapter 5.3, the authors should focus more on if/how small molecules can be envisioned as promoters of miRNA expression and/or activity, rather than describing the miRNAs that are causally linked with disease when depleted.
MINOR
1-Please, go through the text and check for typos. There are a few.
2-Introduction seems a bit patchy. I suggest streamlining it, avoiding the repetition of the same concepts. Also, the focus of the review (targeting miRs with small molecules) should emerge more clearly from the last paragraph.
3-Methods described in 2.1.1-4 are purely in silico and fit as subchapters of the 2.1 Bioinformatic tools chapter. However, Microarray screening is an experimental method and should be presented as chapter 2.2, so that Fluorescence reporter becomes chapter 2.3, and ALIS becomes chapter 2.4.
4-The difference between InfoRNA and DEL methods does not emerge clearly.
5-In Figure 2, panel C is not as intuitive as the others. I suggest to edit it.
6-Table 1/2/3: Legends are not all visible and some images are distorted. Names of the small molecules presented should be added. miRNAs should be listed by increasing number. The title of the first column should be changed into pri/pre-miRs, or in any case it should be stated if the target is the mature miRNA or a precursor. Inhibitors for the same miR should be clustered and presented with the same background color, or eventually 1 background color for each class of inhibitors.
7-In chapter 3.2 (not 3.1) all inhibitors of miR-21 maturation at Dicer level should be described one after the other.
Comments on the Quality of English LanguageEnglish is good, there are only a few typos.
Author Response
Reviewer two
This review article by Tadesse and Benhamou comprehensively describes how small molecules can be used to impair maturation, stability, and function of microRNAs, representing a valuable alternative to “classic” antisense inhibitors.
The review is well conceived and very well written. I commend the authors because it is a pleasure to read it.
Reviewer #2, Comment 1: Please, include in the text a description of the ALIS method (panel G in Fig 2).
Response: We thank the reviewer for his/her comments. Since no research utilized the ALIS method for identifying ligands targeting miRNAs, we have removed Figure G.
Reviewer #2, Comment 2: In chapter 5.3, the authors should focus more on if/how small molecules can be envisioned as promoters of miRNA expression and/or activity, rather than describing the miRNAs that are causally linked with disease when depleted.
Response: We revised the miRNAs focusing points and wrote about how small molecules can be used to enhance the expression of depleted microRNAs. We added two references to explain more about the small molecules used to upregulate the depleted miRs. We added the following detail statement in this chapter:
“For instance, decreased levels of the tumor suppressor miR-107 in plasma are linked to tumor progression and unfavorable outcomes. Reinstating plasma miR-107 levels is suggested as an innovative strategy for treating prostate cancer. Additionally, researchers have identified an intragenic miR, specifically miR-3614, which inhibits the expression of its host gene TRIM25. Notably, IGF2BP3 competes for the binding site, impeding miR-3614 maturation and protecting TRIM25 mRNA from degradation. Elevated levels of miR-3614-3p effectively suppress breast cancer cell growth by reducing TRIM25 expression. Small molecules like Rigosertib are used to inhibit IGF2BP3 which results in decreased TRIM25 levels, suppressed cell proliferation, and shows a collaborative effect with miR-3614-3p overexpression. The let-7 miR family is renowned for its tumor-suppressor role, often downregulated in cancer cells. The Lin28 protein plays a pivotal role by binding to let-7 miR precursors, inhibiting their maturation, and consequently impairing their tumor-suppressive functions. Inhibiting Lin28 protein using small molecules leads to elevated levels of let-7 miR consequently inducing tumor-suppressive functions”.
Reviewer #2, Comment 3: Please, go through the text and check for typos. There are a few.
Response: We corrected the type errors.
Reviewer #2, Comment 4: Introduction seems a bit patchy. I suggest streamlining it, avoiding the repetition of the same concepts. Also, the focus of the review (targeting miRs with small molecules) should emerge more clearly from the last paragraph.
Response: We have made modifications according to the comments to the introduction.
Reviewer #2, Comment 5: Methods described in 2.1.1-4 are purely in silico and fit as subchapters of the 2.1 Bioinformatic tools chapter. However, Microarray screening is an experimental method and should be presented as chapter 2.2, so that Fluorescence reporter becomes chapter 2.3, and ALIS becomes chapter 2.4.
Response: We changed the category of DEL, microarray screening, and fluorescence reporter as independent chapters (2.2, 2.3, 2.4 respectively) separately from the bioinformatic tools. However, we didn’t add the ALIS paragraph since there is no research that utilized ALIS for miR ligand discovery.
Reviewer #2, Comment 6: The difference between InfoRNA and DEL methods does not emerge clearly.
Response: Thanks for highlighting this point it was apparently not clear. DEL is experimental data, in contrast, INFORNA represents computer-based data. Therefore, we included keywords as computer-based and experimental to make it clear to the readers in the manuscript.
Reviewer #2, Comment 7: In Figure 2, panel C is not as intuitive as the others. I suggest to edit it.
Response: We arranged Figure 2C, by removing Figure 2B, about docking to make space for the DEL to enhance clarity. We added the steps and procedures (incubation of target and binding, sequencing, and deconvolution) in the figure.
Reviewer #2, Comment 8: Table 1/2/3: Legends are not all visible and some images are distorted. Names of the small molecules presented should be added. miRNAs should be listed by increasing number. The title of the first column should be changed into pri/pre-miRs, or in any case, it should be stated if the target is the mature miRNA or a precursor. Inhibitors for the same miR should be clustered and presented with the same background color, or eventually 1 background color for each class of inhibitors.
Response: We have arranged the table according to the increasing number of miRs. However, the different color shading that we used is important to show the difference between the small molecules as monomers, homodimers, and hetero-dimers. We used greenish shading to show the homodimer, pink to show the heterodimer, and grey shading to show the monomer ligand. For clarification we added statement to the tables as for table 1 “Monomer molecules are highlighted in grey, homodimers in green, and heterodimers in pink.” Table 2 “Monomers are highlighted in silver, and heterodimer is highlighted in green.” Table 3 “Gray background for RIBOTAC molecules, the blue circles indicate for the RNase L recruiter. Sky blue background for direct degradation by bleomycin (BLM) marked in yellow, while the orange star indicates photosensitizer pyro pheophorbide a (Ppa)”
Reviewer #2, Comment 9: In chapter 3.2 (not 3.1) all inhibitors of miR-21 maturation at the Dicer level should be described one after the other.
Response: We have combined all the miR-21 inhibitors at the top of the table, however in the text we focused on explaining each inhibitor according to the authors rather than the miRs.
Reviewer 3 Report
Comments and Suggestions for Authors
In their review manuscript "Targeting MicroRNAs with Small Molecules," Tadesse et al. timely address the developments in targeting miRNAs by small molecules. Overall, their manuscript is comprehensive and informative, offering a deep dive into the latest advancements in the field. The authors meticulously detail the various strategies employed to target miRNAs, providing valuable insights into the potential therapeutic applications of these small molecules. This review serves as a crucial resource for researchers and clinicians, highlighting the importance of miRNAs in disease pathogenesis and the promising avenues for intervention.
Specific comments and suggestions:
1. Add a table summarizing the miRNAs and their function mentioned in the introduction.
- A table can be added after the introduction to summarize the mentioned miRNAs and their functions. This will provide a quick reference for readers and enhance the clarity of the manuscript.
2. NMR appears only as an acronym.
- It would be beneficial to provide the full form of NMR (Nuclear Magnetic Resonance) the first time it is mentioned in the manuscript to ensure clarity for readers who may not be familiar with the acronym.
3. Is "2.1.4. DNA Encoded Library (DEL)" considered a Bioinformatic tool, or does it need to appear under a different sub-section?
- If the DNA Encoded Library (DEL) is not primarily a bioinformatic tool, it may be more appropriate to place it under a different sub-section that aligns with its primary function or application in the context of miR targeting.
4. In section 2, "Identification of small molecules interacting with miRs," the authors may include a table summarizing the different approaches.
- Including a table to summarize the various approaches for identifying small molecules interacting with miRs can enhance the readability and organization of the manuscript. This table can highlight key features, advantages, and applications of each approach.
5. In section 2, "Identification of small molecules interacting with miRs," the authors may elaborate on the pros and cons of the different methods.
- Expanding on the advantages and limitations of each method will provide a more comprehensive understanding for readers and help them evaluate the suitability of each approach for their research.
6. Discuss limitations such as off-targets, toxicity, and knockdown activity of small molecules in the context of the review.
- Addressing these limitations is crucial for providing a balanced view of the field. The manuscript should include a discussion on the challenges associated with off-target effects, potential toxicity, and the efficiency of knockdown activity of small molecules targeting miRs.
Author Response
Reviewer three
In their review manuscript "Targeting MicroRNAs with Small Molecules," Tadesse et al. timely address the developments in targeting miRNAs by small molecules. Overall, their manuscript is comprehensive and informative, offering a deep dive into the latest advancements in the field. The authors meticulously detail the various strategies employed to target miRNAs, providing valuable insights into the potential therapeutic applications of these small molecules. This review serves as a crucial resource for researchers and clinicians, highlighting the importance of miRNAs in disease pathogenesis and the promising avenues for intervention.
Specific comments and suggestions:
Reviewer #3, Comment 1: Add a table summarizing the miRNAs and their function mentioned in the introduction. A table can be added after the introduction to summarize the mentioned miRNAs and their functions. This will provide a quick reference for readers and enhance the clarity of the manuscript.
Response: We thank the reviewer for his/her comments. In this review article, we are focusing on how small molecules can affect the expression of different over-expressed or depleted microRNAs. We didn’t discuss the miRs that haven’t been targeted by small molecules. However, we discussed and included the miRs that were targeted by small molecules in the three tables categorized by small molecules targeting drosha, dicer, and targeted degradation. Additionally, we have chapter 5.3 which discusses how small molecules can be used to increase the expression of depleted miRs with different mechanisms such as using RNA binding protein inhibitor ligands.
Reviewer #3, Comment 2: NMR appears only as an acronym. It would be beneficial to provide the full form of NMR (Nuclear Magnetic Resonance) the first time it is mentioned in the manuscript to ensure clarity for readers who may not be familiar with the acronym.
Response: We thank the reviewer for his/her comments. We corrected the acronym in the first text and used the second one as NMR.
Reviewer #3, Comment 3: Is "2.1.4. DNA Encoded Library (DEL)" considered a Bioinformatic tool, or does it need to appear under a different sub-section? If the DNA Encoded Library (DEL) is not primarily a bioinformatic tool, it may be more appropriate to place it under a different sub-section that aligns with its primary function or application in the context of miR targeting.
Response: We have changed the chapter numbers and categories accordingly. We changed the category of DEL, microarray screening, and fluorescence reporter as independent chapters (2.2, 2.3, 2.4 respectively) separately from the bioinformatic tools.
Reviewer #3, Comment 4:In section 2, "Identification of small molecules interacting with miRs," the authors may include a table summarizing the different approaches. Including a table to summarize the various approaches for identifying small molecules interacting with miRs can enhance the readability and organization of the manuscript. This table can highlight key features, advantages, and applications of each approach.
Response: The three tables under Drosha, Dicer inhibitors, and degradation include a column about discovery methods for each molecule that is included in the table. This is further discussed in the paragraphs for each methodology as well as in the chapter 3.1, 3.2, 4.1 and 4.2.
Reviewer #3, Comment 5: In section 2, "Identification of small molecules interacting with miRs," the authors may elaborate on the pros and cons of the different methods. Expanding on the advantages and limitations of each method will provide a more comprehensive understanding for readers and help them evaluate the suitability of each approach for their research.
Response: We added sentences that can show the limitations and strengths of each method. Such as for infoRNA, we added a statement at the end of the paragraph. “An inherent drawback of InfoRNA is its reliance on computer-based processes, making it challenging to trust its outputs without the support of experimental validation.” For docking we added “A limitation of molecular docking is its reliance on static structures, which may not fully capture the dynamic nature of RNA structures. MiRs often exhibit conformational changes and flexibility, and accurately predicting these aspects during docking poses challenges, potentially affecting the reliability of the results”. AI statement explaining about its limitation was added in the submitted review. “However, it is important to note that AI-based predictions should be validated experimentally to confirm their accuracy and reliability”. We added the strength and limitation of microarray screening in one statement explained as “Microarray screening provides a powerful means for high-throughput analysis, enabling biomarker discovery and gene expression profiling. However, its utilization is constrained by high costs, data analysis complexity, and the risk of cross-hybridization” In Fluorescence Indicator Displacement (FID), we added “The limitation of FID lies in its dependence on the binding affinity of the indicator molecule, which can potentially hinder the accurate detection of target analytes”. Additionally, we added the limitation of FRET as “FRET offers sensitive, real-time insight into molecular interactions within living cells. However, challenges such as background noise and distance constraints between fluorophores can affect data quality”.
Reviewer #3, Comment 6: Discuss limitations such as off-targets, toxicity, and knockdown activity of small molecules in the context of the review. Addressing these limitations is crucial for providing a balanced view of the field. The manuscript should include a discussion on the challenges associated with off-target effects, potential toxicity, and the efficiency of knockdown activity of small molecules targeting miRs.
Response: In the conclusion, we added a detailed statement about how small molecules can cause off-targets, toxicity, and nonspecific unintended effects and how we can expand the field of study to advance the discovery of precision medicine using various bioinformatic and experimental methodologies. You can find the statement as following in the review. “However, disadvantages include off-target effects due to the miRs' ability to bind to a myriad of mRNA targets, not all of which are necessarily relevant to a specific disease. Additionally, designing small molecules with sufficient affinity and selectivity for specific miRs remains challenging, impeding their clinical translation. Understanding these issues re-quires rigorous computational studies and experimental validation, including in vitro as-says and cellular profiling, to optimize the safety and efficacy of miR-targeting small molecules”.